# Urban–Rural Fringe Long-Term Sequence Monitoring Based on a Comparative Study on DMSP-OLS and NPP-VIIRS Nighttime Light Data: A Case Study of Shenyang, China

**DOI:** 10.3390/ijerph191811835

**Published:** 2022-09-19

**Authors:** Tianyi Zeng, Hong Jin, Zhifei Geng, Zihang Kang, Zichen Zhang

**Affiliations:** 1Key Laboratory of Cold Region Urban and Rural Human Settlement Environment Science and Technology, Ministry of Industry and Information Technology, School of Architecture, Harbin Institute of Technology, Harbin 150001, China; 2Business School, Ningbo University, Ningbo 315211, China

**Keywords:** urban–rural fringe, nighttime light data, dynamic spatial recognition, K-means algorithm, Shenyang

## Abstract

Urban–rural fringes, as special zones where urban and rural areas meet, are the most sensitive areas in the urbanization process. The quantitative identification of urban–rural fringes is the basis for studying the social structure, landscape pattern, and development gradient of fringes, and is also a prerequisite for quantitative analyses of the ecological effects of urbanization. However, few studies have been conducted to compare the identification accuracy of The US Air Force Defence Meteorological Satellite Program’s (DMSP) and the Visible Infrared Imaging Radiometer Suite (VIIRS) nighttime light data from the same year, subsequently enabling long time series monitoring of the urban–rural fringe. Therefore, in this study, taking Shenyang as an example, a K-means algorithm was used to delineate and compare the urban–rural fringe identification results of DMSP and VIIRS nighttime light data for 2013 and analyzed the changes between 2013 and 2020. The results of the study showed a high degree of overlap between the two types of data in 2013, with the overlap accounting for 75% of the VIIRS data identification results. Furthermore, the VIIRS identified more urban and rural details than the DMSP data. The area of the urban–rural fringe in Shenyang increased from 1872 km^2^ to 2537 km^2^, with the growth direction mainly concentrated in the southwest. This study helps to promote the study of urban–rural fringe identification from static identification to dynamic tracking, and from spatial identification to temporal identification. The research results can be applied to the comparative analysis of urban–rural differences and the study of the ecological and environmental effects of urbanization.

## 1. Introduction

The urban–rural fringe is a transitional area between urban and rural areas [1], with obvious transitional characteristics in terms of spatial distribution, economic conditions, demographic characteristics, landscape diversity, and land use. The urban–rural fringe is the frontline area of urban expansion, the area with the most land use problems, and the area with the most traffic and environmental problems [2], which is the weak point of urban planning and management.

Since the 1970s, China’s urban land has expanded at an alarming rate, and the inefficient and disorderly expansion of cities in the new urbanization development plan has led to the urban–rural fringe becoming the weakest part of urban development [3]. Under the urban spatial expansion model, there are differences between the market-driven and community-driven development mechanisms within the urban–rural fringe areas, and they show uneven development in terms of environmental quality, economic competitiveness, and population density, as well as a lack of influence in the urban center, which is not conducive to the formation of a reasonable and healthy urban form [4,5]. Spatial-based analyses of urban fringe zones over a long period of time are helpful in clarifying the integrated development patterns of urban and rural areas, thus addressing the natural, social, and economic problems of urban fringe zones in a targeted manner, assisting in the dynamic monitoring of urban development and policy formulation [6], and providing a reference for decision making in regional integrated spatial development planning.

Scholars usually select one or several indicative elements as the identification elements of urban–rural fringe and analyze their distribution patterns and changing characteristics in order to achieve the identification of urban fringe zones. The identification elements can be extracted based on socio-economic statistics and field survey data, such as population density [7], non-agricultural population ratio [8], economic scale [9], commuting time from urban areas [10], etc. However, due to the lack of continuity of statistical data and the limitation of administrative units it is difficult to reflect the specific differences within administrative areas. Since the identification of urban–rural fringe is mostly carried out at the spatial level, and the spatial data obtained based on remote sensing images and so on can break through the limitation of statistical caliber, therefore, an increasing number of scholars tend to extract the indicative elements of urban fringe based on remote sensing data at multiple spatial and temporal scales, and complete the spatially continuous urban–rural fringe identification based on the spatial grid by setting up a certain scale of spatial grid [11,12].

Nighttime light data can record visible sources of radiation on the Earth’s surface and are a tangible representation of human social activity. The US Air Force Defense Meteorological Satellite Program’s (DMSP) with the Operational Linescan System (OLS) and the National Polar Partnership’s (NPP) with the Visible Infrared Imaging Radiometer Suite (VIIRS) can detect light from human nighttime activities in the wavelength range of 0.5–0.9 μm, including urban nighttime lights and lights from settlements and vehicle traffic [13]. Nighttime light data can be used to characterize population density [14], the spatial distribution of GDP [15], and urbanization processes [16], which provide good entry points for urban–rural fringe identification [17,18]. There are few existing studies that compare DMSP and VIIRS nighttime light data based on dynamic spatial identification of urban–rural fringe [19,20,21]. The main reason for the problem is that the DMSP-OLS ceased to be updated in February 2014, and the NPP-VIIRS continues to capture global nighttime Earth shimmer images [22]. Most of the existing research literature on lighting data is based on the analysis of one of the datasets in isolation, and there is a lack of comparative studies of the two data for the same year to establish the dynamic analysis of long time series. On the one hand, the location of the urban–rural fringe is constantly moving outwards in waves, with the present built-up area being the urban fringe of the past and the present urban fringe being the built-up area of the future [23], making the urban–rural fringe highly dynamic and transient [16]. The monitoring of the morphological characteristics of the urban–rural fringe in time series is of greater theoretical and practical importance than the accuracy of the identification results at a given point in time. On the other hand, based on the same dataset, the identification and dynamic analysis of urban–rural fringe at different points in time can help to clarify the dominant direction of urban expansion and provide guidance for understanding the dynamic process of urbanization in the study area.

Shenyang is the largest city in Northeast China. Since the establishment of the People’s Republic of China, Shenyang has gradually transformed from a national industrial town to a regional center in the northeast [24]. The expansion of the city’s scale and functions have driven the development of the city’s spatial structure, and the expansion and reconfiguration of urban space has provided support for economic development [25]. The analysis of Shenyang’s urban–rural fringe can provide a reference by which to further explore the urban development and transformation of China’s old industrial cities.

Therefore, we took the city of Shenyang as an example and used the K-means algorithm to identify the urban–rural fringe of a long time series based on nighttime light datasets (DMSP-OLS and NPP-VIIRS nighttime lighting image correction and fusion). Since the DMSP-OLS and NPP-VIIRS data overlap in 2012 and 2013, we selected the year 2013 as the starting year and used calibrated DMSP and simulated VIIRS datasets in 2013 for comparison and identification accuracy validation. The data for 2020 were then selected to observe the development of urban–rural fringe in Shenyang. The objectives of the study were: (1) to focus on the two different types of data for better comparison, standardization, accuracy, and possibilities of use; (2) to dynamically track the transformation of Shenyang’s urban–rural fringe zone to clarify the dominant direction and speed scale of Shenyang’s expansion. This paper is the first to compare the identification results of DMSP-OLS and NPP-VIIRS data for the same city in the same year, laying the foundation for future dynamic detection of urban–rural fringe using nighttime lighting data. In addition, the identification and dynamic analysis of urban–rural fringe at different time points are completed based on the optimal results of the cross-sectional comparison.

## 2. Methodology

### 2.1. Introduction to the Study Area

Shenyang city, with a total area of 12,860 km^2^, is located in the southern part of Northeast China, in the middle of Liaoning Province (41°48′11.75″ N, 123°25′31.18″ E), which extends about 125 km from east to west, and about 283 km from north to south. The landscape pattern transitions from low hills in the northeast to undulating sloping plains in front of the mountains, with the broad, flat Lower Liaohe Plain in the central and western parts. Shenyang has a semi-humid continental climate influenced by monsoons in the north temperate zone. The distribution of temperature and precipitation throughout the year decreases from south to northeast and from southeast to northwest.

By the end of 2020, Shenyang will consist of 13 districts (Figure 1) with a resident population of 902.78 million. In Shenyang, there are three functional areas: (1) the core functional area (Heping, Shenhe, Huanggu, and Dadong); (2) the urban development area (Tiexi, Yuhong, Shembei, Sujiatun, Hunnan, and Liaozhong); (3) the surrounding counties and county-level cities (Xinmin, Kangping, and Faku). The division of functional areas is mainly based on historical, economic, demographic, and land price factors. According to the contents of Shenyang’s fourth round of urban master plan (2011–2020): the polycentric pattern of Shenyang’s urban planning has been further deepened. The current central area of Shenyang city is within the Third Ring Road and will be extended to within the Fourth Ring Road in the future. As urbanization continues and the city’s economy develops rapidly, people’s residential trends have begun to shift outwards from the core area, and the Third Ring Road has become an important traffic road in Shenyang’s urban area [26].

With the acceleration of China’s regional integration process in recent years, more and more provinces are hoping to seize the opportunity to compete in the region, and provincial capitals have in recent years incorporated parts of areas outside their municipal boundaries that formerly belonged to other municipalities, or even an entire prefecture-level city, through administrative rezoning, to expand the space for urban development and further enhance the agglomeration effect of economic factors. In 2018, the plenary session of the Liaoning Provincial Party Committee proposed to actively promote the creation of a national central city in Shenyang. With the current GDP and population size of Shenyang, the only way to achieve leapfrog development is to annex the surrounding small cities. The selection of Shenyang as a case study provides a better understanding of the significant transformation of the urban agglomeration development and urban–rural fringe in south-central Liaoning under the administrative restructuring.

### 2.2. Materials

In this study, we primarily used the DMSP/OLS data in 2013 and NPP/VIIRS data in 2020 to identify the urban–rural fringe, both downloaded from NOAA’s National Centers for Environmental Information (NCEI) website [27].

As shown in Table 1, the spatial resolution of DMSP is 1000 m less than the 500 m of VIIRS. The DMSP pixel value has no units and is a discrete grey level value, whereas VIIRS is a continuous radiation value. In addition, DMSP/OLS was originally designed as a meteorological satellite to detect nighttime cloud data and therefore was not equipped with an onboard corrector. When there are no clouds in the sky, urban lighting or Earth shimmer data are captured by the DMSP satellites. Therefore, in order to achieve comparability and continuity of data for 2013 and 2020, both data need to be processed and the NPP/VIIRS data need to be fitted to the DMSP data.

The basic concept was to compress the VIIRS data in the range of 0 to 600 into the same range as that of the DMSP, from 0 to 63, with the same spatial resolution ratio. The main steps include: (1) Referring to the methods of Elvidge et al. [28] and Zhifeng Liu et al. [29], mutual correction and continuity correction of the DMSP-OLS images based on the invariant target area method. The processed data are called calibrated DMSP; (2) According to Kang Wu et al. [30], performing noise reduction in the NPP-VIIRS images to remove extreme outliers and a few negative values caused by gas flares; (3) Referring to Xi Li et al. [13], smooth integration of the two sets of nighttime light data by a power function and Gaussian low-pass filtering. The processed data are then referred to as simulated VIIRS. Figure 2 showed the correlation between the original and simulated datasets in the form of a scatter plot. Clearly, the simulated data were more correlated than the original data. The most widely used scenario for the standardized DMSP/OLS and NPP/VIIRS data is to measure the level of economic activity through its brightness values, and both data sets were found to have very similar historical variation, and there is a high correlation between the two data sets [31,32].

### 2.3. Urban–Rural Fringe Extraction Based on K-Means Algorithm

As a representative of unsupervised clustering algorithms, the main function of the K-means algorithm is to automatically group similar samples into a set, often called “clusters”, with the result that data objects within the same class cluster are as similar as possible, and data objects in different class clusters are as different as possible [33,34]. This property is well suited to image segmentation, where unsupervised machine learning methods can be used without the aid of external information [35], so there is no special requirement for urban morphology in the identification of urban–rural fringe, which has a broader scope of application. Of course, the classification results of K-means can easily fall into the problem of local optimality due to the random selection of initial class cluster centroids [36]. Many studies have examined the impact of urbanization on the urban–rural gradient from different perspectives, identifying that urban and rural areas can be distinguished by features such as landscape character [37,38,39], population density, and local spatial entropy [40], including nighttime lighting.

As urban dwellers have a much higher reliance on nighttime lighting than rural dwellers, the nighttime light intensity and the nighttime light fluctuation in the urban–rural area are also characterized by a transition from urban to rural areas [37]. The light intensity gradually decreases from the city center to the countryside, while the nighttime light fluctuation has a “smooth–fluctuating–smooth” transition. After aligning the pixel values of the DMSP and VIIRS nighttime light images, we extracted two characteristics of nighttime light. The nighttime light intensity was regarded as the digital number (DN) of the composited image. The nighttime light fluctuations reflect the degree of variation in light intensity within a certain range, the index of which was computed from the DN values contained in a certain range. The basic principle is to set the 3 × 3 pixel neighbor, take the difference between its maximum and minimum DN value and generate a new 3 × 3 pixel neighbor, using the Spatial Neighbourhood Analysis tool in ArcGIS 10.5. According to Feng et al. [10], the nighttime light fluctuation index (FI) formula is as in Formula (1):(1)$${\mathrm{FI}}_{n}={\mathrm{DN}}_{\mathrm{nmax}}-{\mathrm{DN}}_{\mathrm{nmin}}$$
where FI_n_ is the degree of nighttime light fluctuation index (FI) in n neighborhood, and DN_nmax_ and DN_nmin_ are the maximum and minimum values of digital number (DN) in the 3 × 3 neighborhood, respectively.

Three regional categories were clustered by K-means using two-dimensional features (DN and FI). The algorithm is based on the idea that k samples are randomly selected as cluster centers from the sample set, and the distance between all samples and these k “cluster centers” is calculated. Then the steps of assigning points and updating cluster centroids are iteratively performed until the change in cluster centroids is minimal or the specified number of iterations is reached [41].

The data sample FI contains n objects FI = {FI_1_, FI_2_, FI_3_,..., FI_n_}, where each object has m (m = 2) dimensional attributes. The goal of the K-means algorithm is to cluster the n objects into specified k (k = 3) class clusters based on the similarity between the objects, with each object belonging to one and only one of the class clusters with the smallest distance to the class center. For K-means, the first step is to initialize the k cluster centers {C_1_, C_2_, C_3_,..., C_n_}, 1 < k ≤ n, and then calculate the Euclidean distance from each object to each cluster center as shown in the following equation:(2)$$\mathrm{dis}\left({\mathrm{FI}}_{i},C_{j}\right)\sqrt{\overset{m}{\underset{t=1}{\sum }}{\left({\mathrm{FI}}_{\mathrm{it}}-C_{\mathrm{jt}}\right)}^{2}}$$

In Equation (2), FI_i_ denotes the i object, 1 ≤ i ≤ n; C_j_ denotes the j cluster center, 1 ≤ j ≤ k; FI_it_ denotes the t attribute of the i object, 1 ≤ t ≤ m; and C_jt_ denotes the t attribute of the j cluster center.

Afterward, the distance of each object to each cluster center is compared in turn, and the object is assigned to the class cluster of the nearest cluster center to obtain k class clusters {S_1_, S_2_, S_3_,…, S_k_}. The K-means algorithm defines the prototype of class clusters in terms of centers, and the class cluster center is the mean of all objects within a class cluster in each dimension, which is calculated as follows [39]:(3)$$C_{l}=\frac{\sum_{X_{i}\in S_{l}}X_{i}}{\left|S_{l}\right|}$$

In Equation (3), C_l_ denotes the center of the l cluster; 1 ≤ l ≤ k denotes the number of objects in the l cluster; and FI_i_ denotes the i object in the lth cluster, 1 ≤ i ≤ |S_l_|.

### 2.4. Verification of Nighttime Light Characteristics and Performance

Sample line

Taking the center of Shenyang’s First Ring Road as the center of Shenyang, a profile line is created running from northwest to southeast throughout the whole city as a sample line. Shenyang’s urban expansion is limited by its administrative boundary, the southern and eastern parts of Shenyang’s urban area had basically developed to the edge of its administrative boundary in 2013. The selection of the sample line from northwest to southeast can clearly show the variation of light intensity and light fluctuation variation characteristics in the three areas. The changes in DN and FI are used to characterize the transition from urban to rural areas.

The probability density distribution

Kernel density is a non-parametric method for estimating the probability density function of a random variable. We use kernel density plots to observe the distribution of DN and FI in urban areas, rural areas, and urban–rural fringe areas, and then evaluate the performance of the combination of nighttime light using the K-means algorithm.

Population density validation

We used county population and total land use data from the China County Statistical Yearbook—County and City Volume and the China County Statistical Yearbook—Township Volume in 2013 and 2020 for subsequent analysis and validation.

The following is the research framework for this paper (Figure 3):

## 3. Comparison of Recognition Results between DMSP Data and Transformed NPP Data in the SAME YEAR

### 3.1. Comparison of Nighttime Light Intensity and Light Fluctuation Characteristics

Table 2 compared the maximum, minimum, mean, and standard deviation of nighttime light intensity and light fluctuations for the two identification results. In terms of general characteristics, the mean values of DN showed a gradient from urban to rural areas, and the standard deviation of DN was highest in the urban–rural combination, indicating that the intensity of light at night in these areas is highly dynamic; as for FI, the minimum values were slightly higher in the urban–rural combination compared to the other areas, but their mean values were much higher than in the other areas.

The nighttime light intensity of the two images generally showed a decreasing trend centered on the core area of the city toward the surrounding countryside (Figure 4a,b). The city area covered the core functional area and most of the expanded urban functional area, and the high DN area broadly matched the contours of the Third Ring Road, except for a slight prominence to the southwest and southeast. Outside the Third Ring Road, light intensity decreases rapidly, with only areas of high values in the surrounding districts and counties. The main difference between the two was that simulated VIIRS has a better capability of observing nighttime lights (Figure 2). The mean DN values for simulated VIIRS were 24.94 and 3.42 for urban–rural fringe and rural areas, which were less than the 28.82 and 6.45 for calibrated DMSP (Table 1). Compared to the calibrated DMSP, the simulated VIIRS has a clearer boundary in the high-value areas, and it observes more of the median nighttime light. The main difference between the two types of images is in Xinmin city and Liaocheng district, where the simulated VIIRS images tend to be brighter and more coherent at the border between Liaocheng district and Tiexi district and between Yuhong district and Xinmin city than the calibrated DMSP.

When it comes to the nighttime light fluctuations, both two images were distributed in a “low–high–low” pattern from the city center outwards, with a clear abrupt change area in the center (Figure 5). The nighttime light fluctuation values reflected a circular profile that conformed better with the ring road of Shenyang than the nighttime light intensity values that demonstrate the city’s peripheral profile. The main difference between the two is that the simulated VIIRS shows a closed loop that is relatively blurred, and there are some contested areas around the closed loop. This is mainly due to the fact that the standard deviation of FI is lower in simulated VIIRS than in calibrated DMSP due to the ability to identify more medium DN values. In urban areas, rural areas, and urban–rural fringe areas, the mean FI values for simulated VIIRS were 7.50, 16.69, and 4.20, respectively, which were slightly lower than the 8.09, 20.27, and 4.63 for calibrated DMSP.

### 3.2. Performance Comparison Combining Nighttime Light Intensity and Light Fluctuation

Firstly, we compared light intensity and light fluctuation variation characteristics by the sample line (Figure 1). Through comparative analysis of the literature and comparison with Google images, we determined the nighttime light thresholds used for determination, and the extracted urban–rural fringe range best matched the actual situation when the DN was less than 15 and the FI was greater than 10. As shown in Figure 6, the light intensity curves of the two images at the sample line were approximately coincident, and although there was some variation in the light fluctuation, it had little effect on the determination of the recognition results. In 2013, the sample points with IDs 1–77 and 140–154 were consistent with the rural area characteristics; the sample points with IDs 77–89 and 121–140 were characterized as urban–rural fringe; and the sample points with IDs 89–140 were typical of urban areas. The above findings indicate that the changing characteristics of nighttime lighting are similar to those of urban land use dynamic attitudes [41,42], which are consistent with the urban–rural structural characteristics. Therefore, the fused nighttime light data can accurately describe the regional structure of the urban–rural interface.

Secondly, we focused on the probability density distribution of DN and FI for the simulated VIIRS data in urban areas, rural areas, and urban–rural fringe areas (Figure 6) and compared the performance of the combination of nighttime light intensity and light fluctuation of the two datasets in 2013, as by analyzing the correlation of multiple dimensions in the data, more information can be obtained [43,44,45].

In regards to nighttime light intensity, as shown in Figure 7a, the probability density for the urban–rural fringes peaked at a DN value of 21. The mean DN value for the urban–rural fringe was between that of the other two types of areas, and the peak probability density was lower than that of the other two types of areas. The nighttime light intensity in the urban–rural fringe had some overlap with the other two areas. For example, the urban areas and the urban–rural fringe had many pixels with a nighttime light intensity very close to each other for DN values in the range of 30–50; there was also some coincidence between rural areas and the urban–rural fringes for DN values in the range of 2–20. Using DN values alone for classification may lead to a large number of misclassifications in the identification of urban–rural fringes. When nighttime light fluctuations were used alone, there was overlap as well (Figure 7b). In 2013, the FI values for the three area types overlapped in the approximate 0–40 range, despite reaching a very low level in the urban areas.

Finally, the two types of data were combined and analyzed simultaneously (Figure 8), it was seen that the three different types of areas could be effectively delineated, meaning that the two datasets can distinguish the urban–rural fringe better by using the two-dimensional characteristics of nighttime lighting data. At the same DN value, simulated VIIRS required a higher FL value to distinguish between two different areas.

### 3.3. Validation of Identification Results

After comparing the performance of nighttime light intensity and light fluctuation, we have listed the combination characteristic of nighttime lights in three different areas and compared the area identified by the two data (Table 3). There was a difference of 439 km^2^ between the two urban–rural fringe recognition areas in 2013. The overlap area was approximately 75% of the simulated VIIRS identification area.

A comparison of the use of calibrated DMSP and simulated VIIRS data to identify urban–rural fringe is shown in Figure 9. There was a significant area of overlap between the results of the two datasets, providing initial confirmation that both methods have a degree of reliability. The area of urban–rural fringe identified by simulated VIIRS is larger compared to calibrated DMSP. The areas with large differences in identification results were mainly in the southwestern part of Xinmin city (Damintun town and Xinglongbao town) and the southwestern part of Kangping County (Dongguan town). Most of them were identified as rural areas when using DMSP data, and mainly as urban–rural fringe when using simulated VIIRS data.

The areas with large differences in identification results were mainly in the southwestern part of Xinmin city (Damintu town and Xinglongbao town) and the southwestern part of Kangping county (Dongguan town) (Figure 10). Most of them were identified as rural areas when using DMSP data, and mainly as urban–rural fringe when using simulated VIIRS data.

Compared to other towns in Xinmin, Damintun town was recognized by the state as a national key town with a high degree of urbanization and relatively good urban infrastructure. Xinglongbao, the most populous town in Xinmin, was a key town in Liaoning Province and a key town in Shenyang for the development of recreation and health. The geographical advantage of being adjacent to the National Highway and close enough to downtown Shenyang made them a priority for development, with a higher degree of urbanization than other towns in Xinmin city and with distinct urban–rural fringe characteristics. From a population perspective, the population density of Damintun town was 282 people/km^2^ and that of Xinglongbao town was 409 people/km^2^, much higher than the average for towns of the same level in Xinmin city (215 people/km^2^) and more than that of Xinmin city’s new town streets (205 people/km^2^). The populations of built-up areas in Damintun and Xinglongbao towns were also the highest among their counterparts in Xinmin, so it can be concluded that these two towns had the characteristics of an urban–rural fringe.

In 2012, Dongguan town was removed by the Chinese government and turned into a street. According to its administrative attributes, most of its area should be classified as an urban–rural fringe. The population density of Dongguan Street is the second highest (205 people/km^2^) among the same level of streets and towns in Kangping County, and the eastern part of Dongguan Street identified by simulated VIIRS can be determined to be an urban–rural fringe. In conclusion, the urban–rural fringe identified by simulated VIIRS data can reveal more details about the urban–rural fringe, and the identification results are more objective and accurate, and more in line with the local development situation. The reason for this is that although the DMSP/OLS satellite’s nighttime bridge crossing time is at 9.30 pm, its satellite sensor deficiency results in the DMSP not being able to capture light in both bright and dim areas, a problem that does not exist in the VIIRS data. This problem is exacerbated by the shortcomings of the DMSP data’s blurred, coarse resolution, resulting in its determination of urban–rural fringe areas concentrated in the brighter lighted ring areas.

## 4. Spatial Expansion of the Urban–Rural Fringe

### 4.1. Urban-Rural Fringe Expansion Analysis

Based on the results of the nighttime lighting data extraction (Figure 9), Shenyang was divided into three parts: urban areas, rural areas, and urban–rural fringe areas. The urban area expanded from within the Third Ring Road to the area within the Fourth Ring Road, and the city expanded extremely rapidly in the southwest, north, and south. From 2013 to 2020, the area of urban–rural fringe in Shenyang grew from 1872 km^2^ to 2537 km^2^, with a growth rate of 35.52%, and the ratio of the total area of Shenyang increased from 14.57% to 19.73%, which was more obvious than the change in urban and rural areas. The urban areas of Shenyang increased from 1399 km^2^ to 1762 km^2^, with a growth rate of 25.95%; the area of rural areas decreased from 8339 km^2^ to 7502 km^2^, with a growth rate of −10.04%. While the Shenyang urban–rural fringe was expanding, its mean DN increased from 24.94 in 2013 to 27.90 in 2020, with a slight decrease in the standard deviation of DN from 9.64 to 9.35 (Table 4). This indicates that the population and economic activities in the urban–rural fringes of Shenyang had strengthened and the internal gap has decreased.

Figure 11 shows Shenyang’s urban and rural spatial patterns in 2013 and 2020. In 2013, the city of Shenyang had shown a tendency to expand to the west and southwest, and there was a trend to connect the urban–rural fringe of the Tiexi and Liaozhong districts, Yuhong district, and Xinmin city. This is because Shenyang does not have the conditions to expand to the east due to the administrative division. With the development of the city, in 2020, the urban–rural fringe had spread across the Tiexi, Yuhong, and Liaozhong districts, with a tendency to connect to the urban area of Xinmin city. In the core functional area and the urban development area, the urban area in 2020 basically covered the urban–rural area in 2013. As time passes, the former urban–rural areas developed into new urban areas. While in the surrounding counties and county-level cities, they developed very slowly. This is also reflected in the change in the administrative division of Liaozhong, which was officially approved by the Chinese State Council in January 2016 for the abolition of the county and the transformation into a district. This helped to expand the city size and enhance the attractiveness of Shenyang as a central city in Northeast China.

### 4.2. Validation of Identification Results in 2020

Compared to 2013, the new urban–rural fringe areas are mainly concentrated in Liaocheng district. By calculating the population density of the county to analyze the actual situation of the towns, it is found that Liaocheng district, Yujafang town, Zhujafang town, Xinmingtun town, Xiaozhaimen town, Changtan town, and Sifangtai town have a population density of more than 300 people/km^2^, all of which rank among the other towns in Xinmin city. In comparison with the population density of 502 people/km^2^ in the built-up area of Xinmin city, combining Google Maps and the local situation, it is possible to identify the area as urban–rural fringe.

We also verified the results of the urban–rural fringe identification by comparing the distribution of the POI data and Google Earth images. The POI (Point of Interest) is derived from the vector dataset of point-like map elements in DLG (Digital Line Graphic) products of basic mapping results [46], which can be abstracted into points for management, analysis, and calculation in GIS (Geographic Information System) [47]. Compared with the method of verifying only by urban population, POI can better reflect the actual location of people’s activities, cross administrative boundaries, and reflect differences within administrative districts [48]. As can be seen from Figure 12, the POI distribution of the discrepancy area roughly summarizes the contours of the area. Without considering the POI of the urban center, the POI of the urban–rural interface in the townships where the different areas are located accounts for roughly 98% of the POI, indicating the high accuracy of this study.

## 5. Limitations and Research Perspectives

Firstly, in terms of the drawbacks of the K-means algorithm, the final result generated often depends heavily on the location of the centroids at the beginning. As the K-means algorithm is inherently iterative, the clustering results tend to converge on a local optimum and not a global optimum [49]. In addition, the clustering is susceptible to isolated points. K-means clusters are calculated based on all sample data in the cluster, and isolated objects can bias the determination of the clusters, potentially affecting the stability and accuracy of the clustering [50].

Secondly, although we carried out a dynamic analysis of long time series data, we found that the DMSP data of nighttime lights are of low resolution and it was difficult to identify small-scale rural lights; thus, the rural light data had a DN and FI of 0. The combination of Landsat 8, SPOT (Systeme Probatoire d’Observation de la Terre), and other higher-resolution data for the spatial identification of the urban–rural interface is worth exploring [19], and this is something that needs to be further developed in the future.

Thirdly, the diversity of the urban–rural fringe in terms of its morphology and function makes it a multifaceted socio-economic body that cannot be easily classified by socio-economic or strict spatial criteria alone [51]. Therefore, the integrated application of data for the identification of urban fringe zones may become an important research direction in the future for the analysis of differences in the identification results caused by different identification elements and for the determination and selection of the most suitable identification elements according to different research objectives so as to achieve the advantages of multiple, complementary identification results.

Finally, we only explored the possibility of the dynamic identification of the urban–rural fringe, but the evolution of the urban–rural fringe itself is also highly scale-dependent, with different processes acting at different frequencies and spatial scales, and their cyclical patterns and dynamic characteristics, which are also the dynamics of the urban–rural fringe, are not always the same. An inadequate choice of scale can lead to misconceptions about the scientific nature of the object of study [52]. Specifically, depending on the purpose of identification and the differences between cities, different sizes of spatial grids can be used as identification units in future urban–rural fringe identification research, and different time intervals can be considered to discuss the differences and similarities in the spatial identification results and temporal dynamics.

## 6. Conclusions

Long time series monitoring of the urban–rural fringe is a prerequisite for the quantitative analysis of the ecological effects of urbanization. Previous studies of urban–rural fringe identification based on nighttime lighting data used only DMSP alone, with few comparative studies of DMSP and VIIRS, neglecting the dynamic monitoring of urban–rural fringe in specific cities. This paper proposes a new framework to compare the identification results using two types of nighttime lighting data. This is used as a basis for urban–rural fringe identification and dynamic analysis at different points in time, based on the same set of data.

We validated the possibility of dynamically tracking changes in urban–rural fringe based on the same set of nighttime lighting data by comparing the results of urban–rural fringe identification from calibrated DMSP and simulated VIIRS in 2013 to deal with environmental issues in innovative ways. The dynamic changes of urban–rural fringe in Shenyang in 2013 and 2020 were also analyzed, completing a quantitative analysis of the urban–rural fringe expansion in the study area over a long time series. The results show that the restructuring of the administrative division has had a significant impact on the expansion of the city, with the urban–rural fringe in the central district of Shenyang continuously expanding towards the southwest and joining the urban–rural fringe in the original Liaocheng district. The results of the study help to summarize urban expansion patterns and provide an important basis for the comparative analysis of urban–rural differences and the study of the ecological effects of urbanization. The results show that calibrated DMSP and simulated VIIRS have a high degree of overlap, but simulated VIIRS has a higher accuracy rate in the identification of some towns. The results of the study will help to improve the effective identification of urban and rural spatial patterns, and effectively serve urban spatial design, land use planning, and environmental protection planning.

## Figures and Tables

**Figure 1 ijerph-19-11835-f001:**
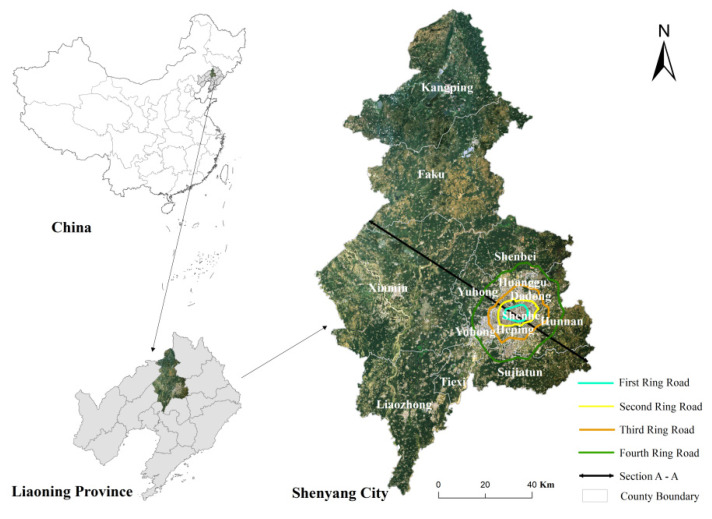
Location and ring roads in Shenyang.

**Figure 2 ijerph-19-11835-f002:**
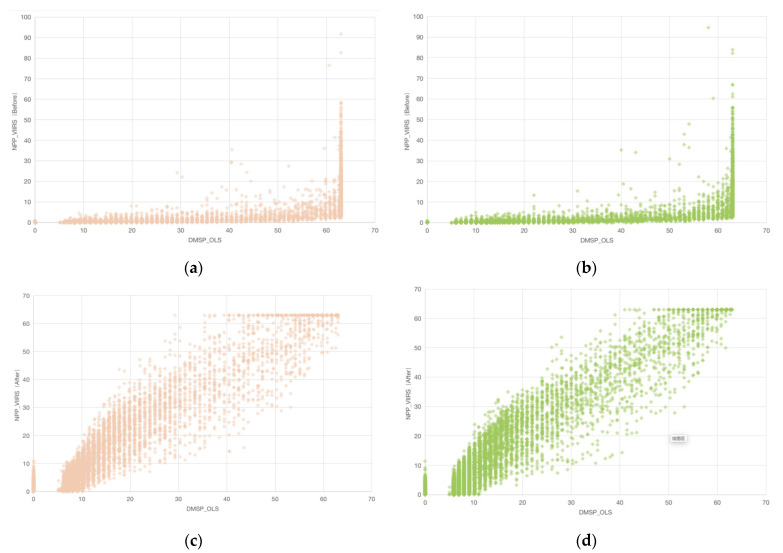
Distribution of pixel values of original DMSP and VIIRS annual composites in 2012 (**a**) and 2013 (**b**) in Shenyang, and original DMSP and simulated VIIRS annual composites in 2012 (**c**) and 2013 (**d**) in Shenyang.

**Figure 3 ijerph-19-11835-f003:**
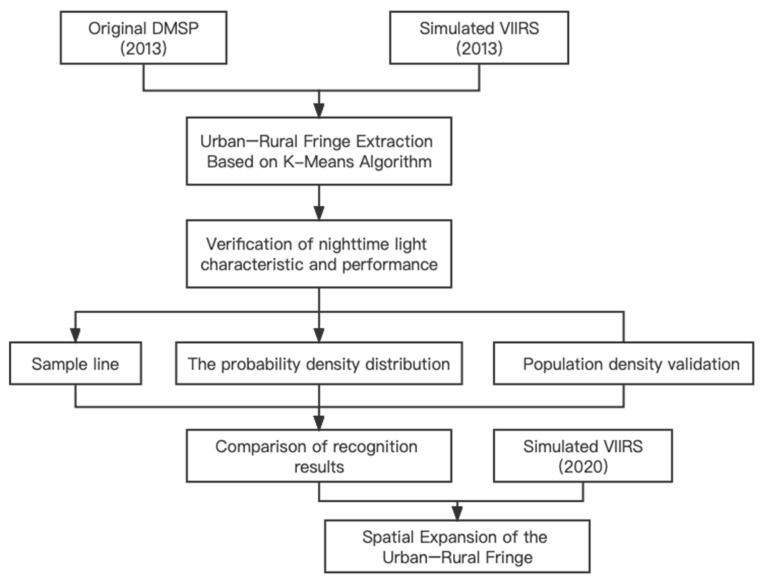
The research framework of this paper.

**Figure 4 ijerph-19-11835-f004:**
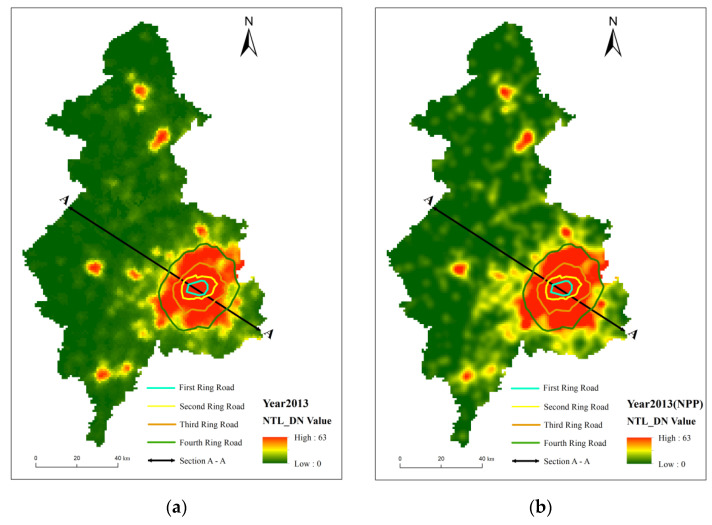
Spatial distribution of nighttime light digital number in Shenyang in 2013 (**a**) and 2020 (**b**).

**Figure 5 ijerph-19-11835-f005:**
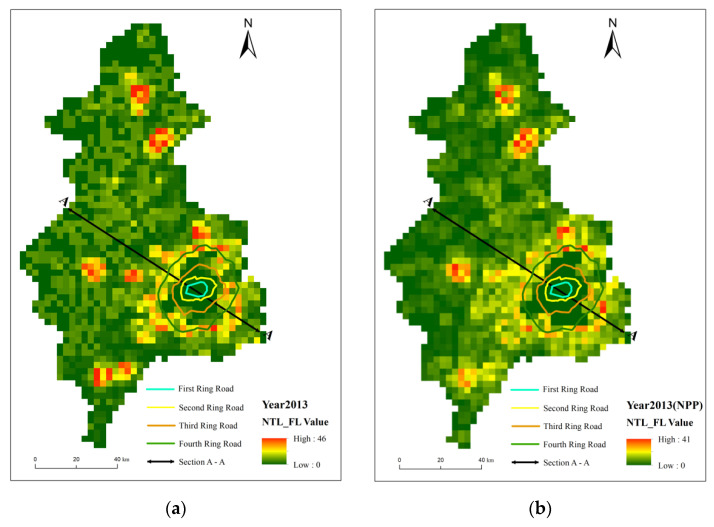
Spatial distribution of light fluctuation index in Shenyang in 2013 (**a**) and 2020 (**b**).

**Figure 6 ijerph-19-11835-f006:**
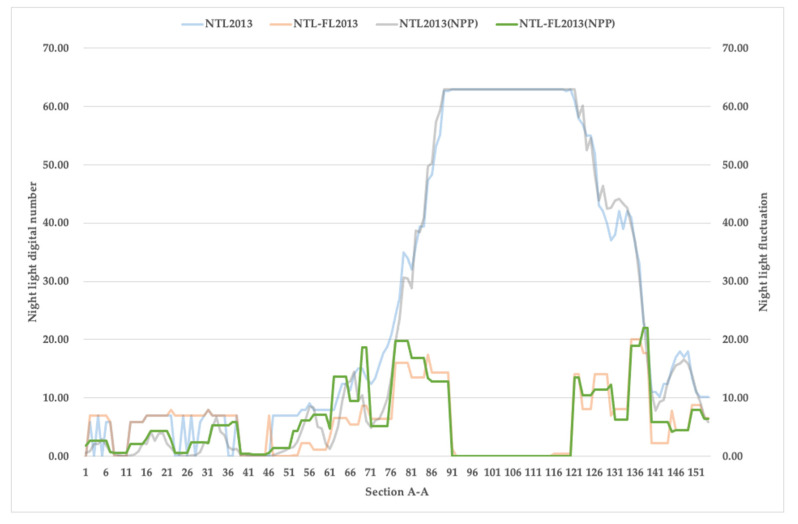
The cross-sectional view of the nighttime light intensity and the nighttime light fluctuation.

**Figure 7 ijerph-19-11835-f007:**
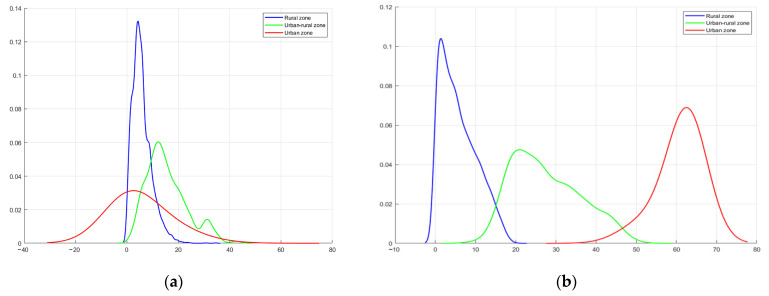
The probability density distribution of DN (**a**) and FI (**b**) of simulated VIIRS for the three regional types. The *x*-axis in (**a**) is the nighttime light digital number; the *x*-axis in (**b**) is the nighttime light fluctuation; and the *y*-axis in (**a**,**b**) is the distribution frequencies.

**Figure 8 ijerph-19-11835-f008:**
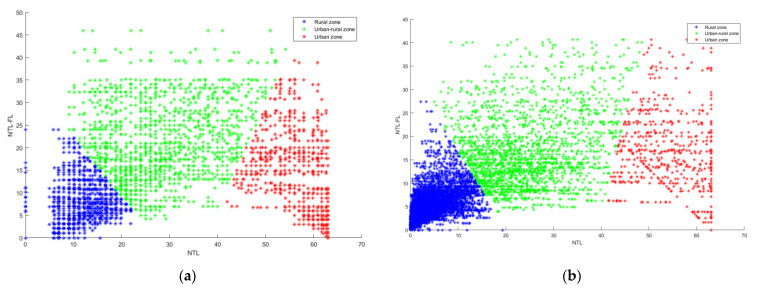
The distribution of DN and FI of calibrated DMSP (**a**) and simulated VIIRS (**b**) for the three regional types.

**Figure 9 ijerph-19-11835-f009:**
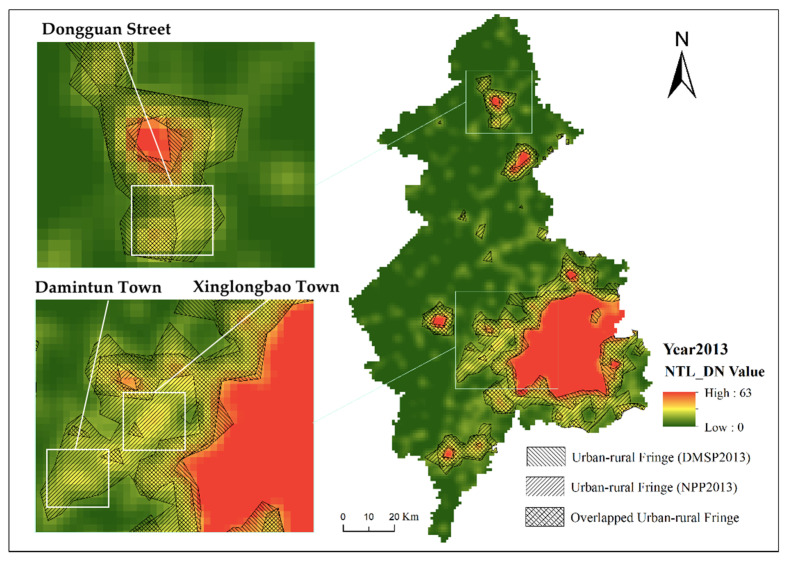
Comparison of the urban–rural fringes identified based on calibrated DMSP and simulated VIIRS.

**Figure 10 ijerph-19-11835-f010:**
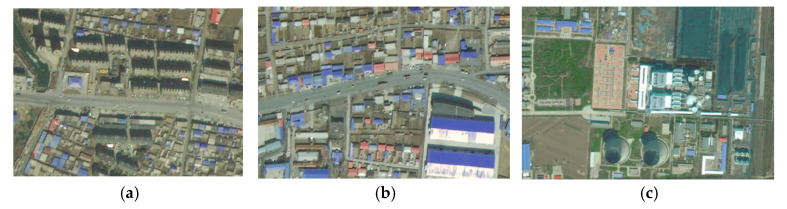
Aerial photo of Damintun town (**a**), Xinglonbao town (**b**), and Dongguan Street (**c**).

**Figure 11 ijerph-19-11835-f011:**
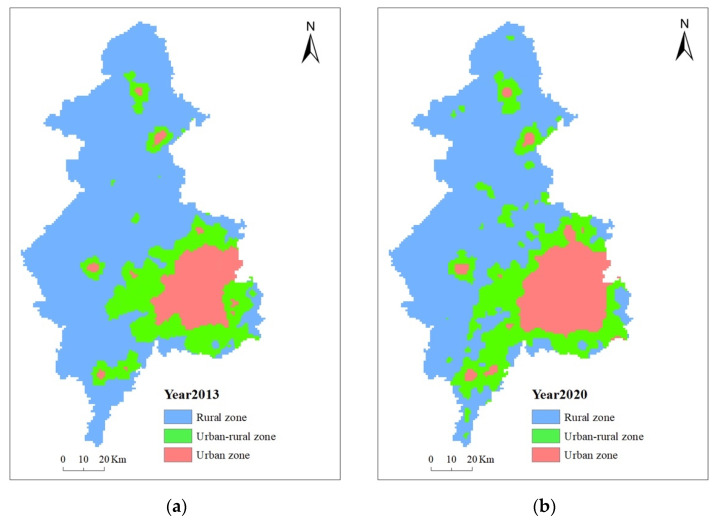
The spatial distribution pattern of urban–rural fringes in 2013 (**a**) and 2020 (**b**).

**Figure 12 ijerph-19-11835-f012:**
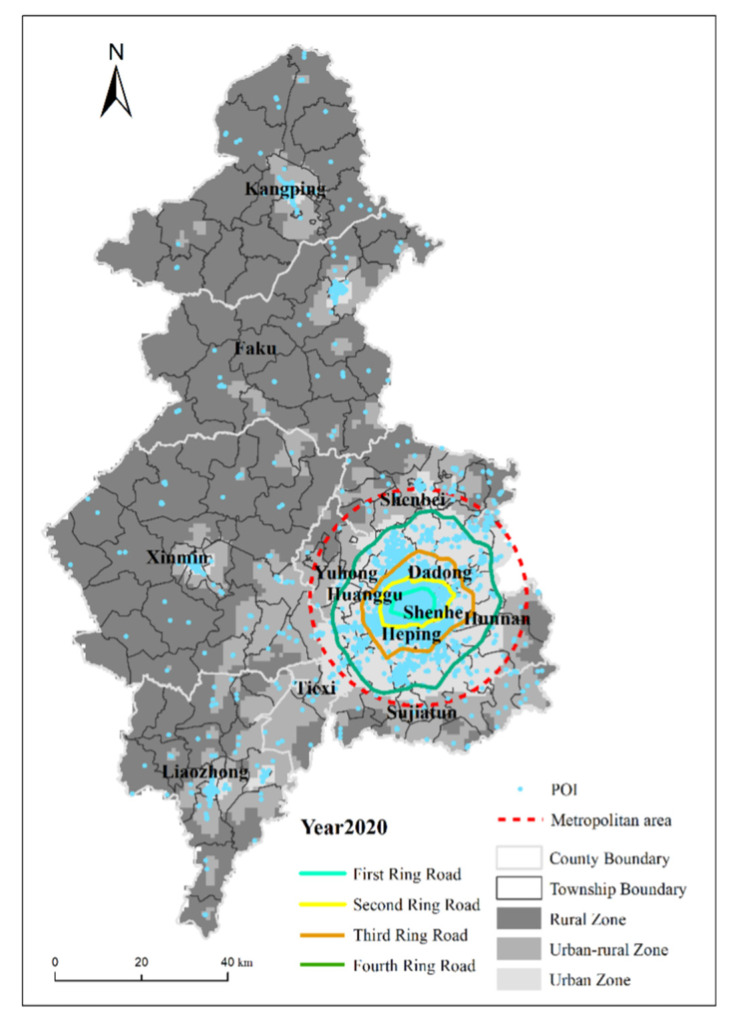
POI test for spatial identification results of urban–rural fringes.

**Table 1 ijerph-19-11835-t001:** Comparison of DMSP/OLS and NPP/VIIRS annual data characteristics.

Source	DMSP/OLS	NPP/VIIRS
Spatial resolution	1000 m	500 m
Onboard calibration	No	Yes
Units of pixel values	Relative	Radiance (nanoWatts/(cm^2^ sr))
Available temporal sequence	1992–2013 annual composites	2012–present monthly composites
Range of pixel values	0–63	0–472.68 ^1^

^1^ This range of values is our processed range; the actual range of values is larger than this.

**Table 2 ijerph-19-11835-t002:** Comparison of two datasets nighttime light intensity and light fluctuation characteristics in 2013.

	Urban Area		Urban–Rural Fringe		Rural Area	
	DMSP	VIIRS	DMSP	VIIRS	DMSP	VIIRS
Min DN	29.00	33.01	7.99	3.31	0.00	0.00
Max DN	63.00	63.00	60.78	62.54	28.23	21.37
Mean DN	59.11	59.56	28.82	24.94	6.45	3.42
Standard deviation of DN	5.67	6.25	10.59	9.64	5.09	3.86
Min FI	0.00	0.00	1.11	2.61	0.00	0.00
Max FI	46.00	40.67	46.00	40.67	46.00	40.07
Mean FI	8.09	7.50	20.27	16.69	4.63	4.20
Standard deviation of FI	9.31	9.75	8.83	7.72	4.16	3.84

**Table 3 ijerph-19-11835-t003:** Area and nighttime light of identified regions in Shenyang city.

	Urban Area		Urban–Rural Fringe		Rural Area	
	DMSP	VIIRS	DMSP	VIIRS	DMSP	VIIRS
Area (km^2^)	1257	1399	1433	1872	9111	8339
Light Intensity	High		Middle		Low	
Light Fluctuation	Low		High		Low	
Combination Characteristic	High–Low		Middle–High		Low–Low	

**Table 4 ijerph-19-11835-t004:** Area and nighttime light of identified regions in Shenyang city in 2013 and 2020.

	Urban Area		Urban–Rural Fringe		Rural Area	
	2013	2020	2013	2020	2013	2020
Area (km^2^)	1399	1762	1872	2537	8339	7502
Min DN	33.01	37.72	3.31	5.06	0.00	0.00
Max DN	63.00	63.00	62.54	63.00	21.37	25.70
Mean DN	59.56	60.71	24.94	27.90	3.42	6.02
Standard deviation of DN	6.25	4.96	9.64	9.35	3.86	4.71
Min FI	0.00	0.00	2.61	1.48	0.00	0.00
Max FI	40.67	43.75	40.67	43.75	40.07	43.75
Mean FI	7.50	5.30	16.69	15.15	4.20	6.02
Standard deviation of FI	9.75	8.29	7.72	7.90	3.84	4.14

## Data Availability

Not applicable.

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
