# Peer review of "Urban–Rural Fringe Long-Term Sequence Monitoring Based on a Comparative Study on DMSP-OLS and NPP-VIIRS Nighttime Light Data: A Case Study of Shenyang, China"

_ijerph, 2022, doi:10.3390/ijerph191811835_

Round 1

Reviewer 1 Report

Dear authors thank you for choosing this journal for your study.
The topic turns out to be very interesting and the method of study conducted turns out to be original with good future prospects. Within this framework we suggest extending the conclusions by better specifying the future prospects of the study ( strengths and weaknesses of the method).
It would also be optimal to be able to have more connection between the simulations and the real situation of the areas so in Figure 8, for "Dongguan street and Damintun Town/Xinglonbao Town" I would suggest putting real photographs of the areas, to show the reader, the real state of the areas (aerial photo and street level photo ).
Also I would suggest expanding the bibliography with some more international references (it is all very focused on Chinese studies) this is to give an even broader dimension to the study.

Reviewer 2 Report

This manuscript focuses on a topic of sure interest to the readership of the International Journal of Environmental Research and Public Health. However, it has some unclear issues. My main concerns are the following:

- The title of the manuscript should include the geographical location of the case study area.

- The introduction could be clearer. An extensive literature review should be included to understand what has been researching in the field and help to identify research gaps for your research. In addition, in the introduction section, the authors should indicate what are the innovative contributions of your manuscript to science.

- Please include a methodological framework in the materials and methods section.

- Case study contextualization is not enough. I believe one of the aims of the work is to make readers understand the magnitude of the issue authors are debating; and why the authors choose this specific case study? This should be well justified.

- The methodology needs more explanations regarding alternative approaches.

- The discussion is missing. The authors should compare your results with previous studies and explain why your results are similar or different from previous findings.

- Please emphasise the contribution and implication of the paper. There should be at least a clear contribution to the field of knowledge. This manuscript should generate some kind of contribution to the scientific field (e.g., the proposal of a new framework, an innovative approach to a methodology, etc.).

- In the conclusion section the authors stated that “The results of the study will help to improve the effective identification of urban and rural spatial patterns, and effectively serve urban spatial design, land use planning, and environmental protection planning.” But can the authors recommend how the results achieved in this research can support decision-makers coordinate spatial planning policies? Or any proposals on how these results can be applied to improve mechanisms for the implementation of spatial policies?

- Minor grammar and punctuation errors can be found throughout the text and need to be corrected.

Round 2

Reviewer 2 Report

Thank you for your revisions to the manuscript. It has improved a lot. However, before publication, I will suggest not using acronyms in the title.

Author Response

Thank you so much for your careful check. After referring to relevant papers, we have changed the title to "Urban-Rural Fringe Long-Term Sequence Monitoring Based on a Comparative Study on DMSP-OLS and NPP-VIIRS Nighttime Light Data: A Case Study of Shenyang, China"